# Uniform-Precision Neural Network Quantization via Neural Channel Expansion

## Abstract

Uniform-precision neural network quantization has gained popularity thanks to its simple arithmetic unit densely packed for high computing capability. However, it ignores heterogeneous sensitivity to the impact of quantization across the layers, resulting in sub-optimal inference accuracy. This work proposes a novel approach to adjust the network structure to alleviate the impact of uniform-precision quantization. The proposed neural architecture search selectively expands channels for the quantization sensitive layers while satisfying hardware constraints (e.g., FLOPs). We provide substantial insights and empirical evidence that the proposed search method called *neural channel expansion* can adapt several popular networks' channels to achieve superior 2-bit quantization accuracy on CIFAR10 and ImageNet. In particular, we demonstrate the best-to-date Top-1/Top-5 accuracy for 2-bit ResNet50 with smaller FLOPs and the parameter size.

## 1 Introduction

Deep neural networks (DNNs) have reached human-level performance in a wide range of domains including image processing (He et al. (2016); Tan & Le (2019)), object detection (Ren et al. (2015); Liu et al. (2016); Tan et al. (2020)), machine translation (Wu et al. (2016); Devlin et al. (2018)), and speech recognition (Zhang et al. (2016); Nassif et al. (2019)). However, tremendous computation and memory costs of these state-of-the-art DNNs make them challenging to deploy on resource-constrained devices such as mobile phones, edge sensors, and drones. Therefore, several edge hardware accelerators specifically optimized for intensive DNN computation have emerged, including Google's edge TPU(Google (2019)) and NVIDIA's NVDLA (NVIDIA (2019)).

One of the central techniques innovating these edge DNN accelerators is the quantization of deep neural networks (QDNN). QDNN reduces the complexity of DNN computation by quantizing network weights and activations to low-bit precision. Since the area and energy consumption of the multiply-accumulate (MAC) unit can be significantly reduced with the bit-width reduction (Sze et al. (2017)), thousands of them can be packed in a small area. Therefore, the popular edge DNN accelerators are equipped with densely integrated MAC arrays to boost their performance in compute-intensive operations such as matrix multiplication (MatMul) and convolution (Conv).

Early studies of QDNN focused on the quantization of weights and activations of MatMul and Conv to the same bit-width (Hubara et al. (2016); Rastegari et al. (2016); Zhou et al. (2016)). This uniform-precision QDNN gained popularity because it simplifies the dense MAC array design for edge DNN accelerators. However, uniform bit allocation did not account for the properties of individual layers in a network. Sakr & Shanbhag (2018) showed that the optimal bit-precision varies within a neural network from layer to layer. As a result, uniform-precision quantization may lead to sub-optimal inference accuracy for a given network.

Mixed-precision networks address this limitation by optimizing bit-widths at each layer. In this approach, the sensitivity of the layer to the quantization error is either numerically estimated (Zhou et al. (2017); Dong et al. (2019)) or automatically explored under the framework of neural architecture search (NAS, Wang et al. (2019); Elthakeb et al. (2018)) to allocate bit-precision properly. However, mixed-precision representation requires specific variable precision support in hardware, restricting computation units' density and power efficiency (Camus et al. (2019)). Therefore, mixed-precision support imposes a significant barrier for the low-profile edge accelerators with stringent hardware constraints.

In this work, we propose a novel NAS based hardware-friendly DNN quantization method that can address the layer-wise heterogeneous sensitivity under uniform-precision quantization. The proposed method explores network structure in terms of the number of channels. Different from the previous work that only includes pruning of the channels in its search space (Dong & Yang (2019)), we further incorporate the expansion of the channels, thus called *neural channel expansion* (NCE). During a search of NCE, search parameters associated with different numbers of channels are updated based on each layer's sensitivity to the uniform-precision quantization and the hardware constraints; the more sensitive to quantization errors, the larger number of channels preferred in that layer. When the preference to the larger number of channels in a layer exceeds a certain threshold, we expand the channels in that layer's search space so that the more number of channels can be explored. Therefore, NCE allows both pruning and expansion of each layer's channels, finding the sweet-spot for the trade-off between the robustness against the quantization error and the hardware cost. We analytically and empirically demonstrate that NCE can adequately facilitate the search to adapt the target model's structure for better quantization accuracy. The experimental results on CIFAR10 and ImageNet show that the network structures adapted from the popular convolutional neural networks (CNNs) achieve superior accuracy when the challenging 2-bit quantization is uniformly applied to MatMul and Conv layers. In particular, we achieve the best-to-date accuracy of 74.03/91.63% (Top-1/Top-5) for NCE-ResNet50 on ImageNet with slightly lower FLOPs and 30% reduced number of parameters.

Our contributions can be summarized as follows:

- We propose a new NAS-based quantization algorithm called *neural channel expansion* (NCE), which is equipped with a simple yet innovative channel expansion mechanism to balance the number of channels across the layers under uniform-precision quantization.
- We provide an in-depth analysis of NCE, shedding light on understanding the impact of channel expansion for compensation of quantization errors.
- We demonstrate that the proposed method can adapt the structure of target neural networks to significantly improve the quantization accuracy.

## 2 RELATED WORK

**Neural architecture search**: The goal of NAS is to find a network architecture that can achieve the best test accuracy. Early studies (Zoph & Le (2016); Zoph et al. (2018)) often employed meta-learners such as reinforcement learning (RL) agents to learn the policy for accurate network architectures. However, RL-based approaches may incur prohibitive search costs (e.g., thousands of GPU hours). As a relaxation, differentiable neural architecture search (DNAS) has been proposed (Liu et al. (2018)), which updates the search parameters and the weights via bi-level optimization. Recent DNAS approaches considered hardware constraints such as latency, the number of parameters, and FLOPs so that the search explored the trade-off between the cross-entropy loss and the hardware constraint loss. This search resulted in the discovery of light-weight models. As an example, Dong & Yang (2019) explored channel pruning that satisfies the target hardware constraints. In this work, we adopt the successful NAS framework in the domain of QDNN, for which we devise a novel channel expansion search to robustify networks against the quantization errors.

**Low-precision quantization of deep neural network**: QDNN has been actively studied in the literature. Early work on QDNN (Hubara et al. (2016); Rastegari et al. (2016); Zhou et al. (2016)) introduced the concept of a straight-through estimator (STE) for the approximation of gradients of the non-differentiable rounding operation. This approximation enabled uniform-precision (1- or multi-bit) quantization during the model training procedure, which fine-tunes the weight parameters towards lower training loss. QDNN techniques have evolved to adaptively find the quantization step size (Choi et al. (2018); Zhang et al. (2018); Jung et al. (2019); Esser et al. (2020)), which significantly enhanced the accuracy of the uniform-precision quantization. However, this line of research lacks consideration of the heterogeneous quantization sensitivity for individual layers in a network. On the other hand, mixed-precision quantization allows layer-specific bit-precision optimization; the higher bit-precision is assigned to the more quantization sensitive layers. Zhou et al. (2017); Dong et al. (2019) numerically estimated the sensitivity via approximating the impact of quantization errors on model prediction accuracy. Wang et al. (2019); Elthakeb et al. (2018) employed a reinforcement learning framework to learn the bit-allocation policy. Wu et al. (2018); Cai & Vasconcelos (2020) adopted DNAS with the various bit-precision operators in the search space. However, mixed-precision

representation requires specific variable precision support in hardware, restricting computation units' density and power efficiency (Camus et al. (2019)). Therefore, mixed-precision support imposes a significant barrier for the low-profile edge accelerators with stringent hardware constraints. In this work, we exploit NAS to address the layer-wise heterogeneous sensitivity under uniform-precision quantization.

**Channel expansion for accurate DNN quantization**: Researchers have actively studied channel expansion for accurate DNN quantization. Pioneering work by Mishra et al. (2018) (WRPN) demonstrated that an increased number of channels during the training helped regain QDNN accuracy, but this work lack discussion about the detailed mechanism of channel expansion compensating the quantization error. Zhao et al. (2019) and Park & Choi (2019) further attempted to split the channels with large magnitude weights in the pre-trained models. This channel splitting reduced the dynamic range of weights to be represented with lower bit-precision (6-8-bits). Regarding the control over the dynamic range, Meller et al. (2019) also adjusted the scale factors of the weight parameters after training to balance the dynamic range across the layers. However, these approaches focused on the numerical remedy for quantization of pre-trained models (with relatively high bit-precision). Thus, it not straightforward to extend their work for quantization-aware training, which is necessary for ultra-low bit QDNN. We provide insights with empirical supports that the structure with the channel expanded layers itself matters, reducing the dynamic range of activation during the quantization-aware training. These exciting insights become a pivotal motivation for us to explore channel expansion in the NAS framework.

## 3 NEURAL CHANNEL EXPANSION

In this section, we explain the detail of our neural channel expansion method. Similar to TAS (Dong & Yang (2019)), we construct the search space over the number of channels $C = \{1 : c_{out}\}$ with the search parameters $\alpha \in \mathbb{R}^{|C|}$. Then the output activation is computed as the weighted sum of sampled activations with a different number of channels aligned via channel-wise interpolation (CWI):

$$\hat{O} = \sum_{j \in I} Softmax(\alpha_j; \{\alpha_k\}_{k \in I}) \times CWI(O_{1:C_j}, max\{c_{out}^k\}_{k \in I}), \qquad (1)$$

where output activation $O_{j:1 \leq j \leq c_{out}} = \sum_{k=1}^{c_{in}} Q(X\{k, :, :\}) * Q(W\{j, k, :, :\})$ is computed with input activation $X$ and weight parameters $W$ quantized by the quantizer $Q$, and $I$ is the sampled subset of $C$.

During the search, the search parameters are updated via channel selection based on the trade-off between the cross-entropy loss and the hardware constraint loss (e.g., FLOPs). In TAS, the number of channels ($|C|$) is fixed, limiting the exploration scope to the pruning. In NCE, we enable channel expansion of individual layers when the search parameter associated with the maximum number of channels exceeds a certain threshold. The intuition is that if one layer is susceptible to the quantization errors, its search parameters are updated toward the preference for a larger number of channels to decrease the cross-entropy loss. With this simple expansion condition, we can expand channels to those layers affected most by the quantization errors and prune channels of the other layers robust to quantization; therefore, the overall hardware constraints are met.

Algorithm 1 summarizes the overall procedure. NCE consists of three phases: warm-up, search, and train. As advocated by Wu et al. (2018) and Bender et al. (2020), we first perform a warm-up of the entire super-net so that all the super-net weight parameters can be reasonably initialized. The search phase consists of the iterative updates of weights ($w$) and the search parameters ($\alpha$) via bi-level optimization. The updated search parameter associated with the maximum number of channels is compared with a threshold (pre-determined as a hyper-parameter) to identify if a layer needs a channel expansion in each layer. When a channel expansion happens (= *Expand*) , the additional weight parameters are added to that layer (and the search parameter is also copied), increasing the number of channels. Once the search is done, the candidate model is derived by the "winner-takes-all" strategy; i.e., for each layer, the number of channels with the largest magnitude search parameter is selected.

---

**Algorithm 1: Neural Channel Expansion**

---

**Input:**
 Split the training set into two dis-joint sets: $D_{weight}$ and $D_{arch}$ $(n(D_{weight}) = n(D_{arch}))$
 Search Parameter: $\{\alpha_1^l, \alpha_2^l, .., \alpha_n^l\} \in A^l$, $\{A^1, A^2, .., A^L\} \subset \mathbb{A}$, $L$ =number of layer
 Expand Threshold: $T$

1 **For** Warm-up Epoch **do**
2  Sample batch data $D_w$ from $D_{weight}$ and network from $\mathbb{A} \sim U(0, 1)$
3  Calculate $Loss_{weight}$ on $D_w$ to update network weights
4 **End For**
5 **For** Search Epoch **do**
6  Sample batch data $D_w$ from $D_{weight}$ and network from $Softmax(\mathbb{A})$
7  Calculate $Loss_{weight}$ on $D_w$ to update network weights
8  Sample batch data $D_a$ from $D_{arch}$ and network from $Softmax(\mathbb{A})$
9  Calculate $Loss_{arch}$ on $D_a$ to update $\mathbb{A}$
10  **For** layer **do**
11   $j \leftarrow \#A^l$
12   **If** $Softmax(\alpha_j^l; \{\alpha_k^l\}_{k \in j}) \geq T$ **do**
13    *Expand* search space$(\alpha_{j+1}^l)$
14    $\alpha_{j+1}^l \leftarrow \alpha_j^l$  **# copy search parameter**
15   **End if**
16  **End for**
17 **End for**
18 Derive the searched network from $\mathbb{A}$
19 Randomly initialize the searched network and optimize it on the training set

---

## 4 ANALYSIS

This section explains how NCE finds the network structures that are more robust to the uniform-precision quantization error while maintaining hardware constraints. We first reveal insightful observations that the expanded channel's structure reduces the input activation's dynamic range and suppresses quantization errors. In other words, a network structure adapted by NCE can be trained standalone from scratch and exhibit robustness to the quantization. This finding motivates us to seek a NAS based exploration for channel expansion; we show that NCE can facilitate compensation of the quantization errors by selective channel expansion.

### 4.1 IMPACT OF CHANNEL EXPANSION ON DYNAMIC RANGE OF ACTIVATION

As discussed in Sec.2, it is well studied that channel-splitting can decrease the dynamic range of the weight parameters. However, it is not clear how much its structure itself affects quantization during neural network training. To understand the impact of expanded channel structure on DNN quantization, we first show that quantization applied to a given network substantially increases the dynamic range of activation, hindering successful QDNN. Fig.1a shows the standard-deviation (STDEV) of input activation for ResNet20 trained from scratch on the CIFAR10 dataset, with and without quantization during training[1]. W{X}A{Y} indicates that weights and activations are quantized into X- and Y-bits, respectively. The substantial increase of STDEV for W32A2 and W2A2 implies that large quantization errors would occur when input activation is quantized. (See Appendix for detail discussion on the quantization error.) Such an increase of dynamic range could partially explain why quantization with the fixed model structure often suffers significant accuracy degradation when ultra-low bit-precision is uniformly applied. This intriguing phenomenon can be observed in all the models we investigated (e.g., VGG16 in Fig.1c).

In addition, Fig. 1b and c show the impact of expanded channel structure on the dynamic range. "2X" models indicate that the number of channels in their layers is doubled. All the hyper-parameter

---

[1]This quantization-aware training follows the same hyper-parameter settings described in Sec.5.1.

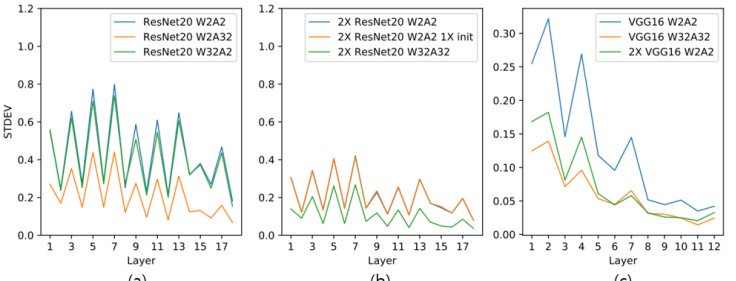

Figure 1: (a-b) Layer-wise STDEV for ResNet20 with 1X or 2X channels. (c) Layer-wise STDEV for VGG16 with 1X or 2X channels. (d) Test accuracy on ResNet20 with 1X or 2X channels (accuracy drop from W32A32 to W2A2).

settings are the same. Notice that both 2X ResNet20 and VGG16 models with 2-bit quantization (= W2A2) could reduce the STDEV down to the full-precision 1X models.

One might regard this dynamic range reduction due to weight initialization with 2X channels, based on the idea similar to channel splitting discussed in Sec.2. However, we examined that the initial weight parameters have little impact on the input activation's dynamic range. More specifically, since we used "He" initialization (He et al. (2015))[2], the number of channels determines the dynamic range of the initial weights. As shown in Fig.1a, however, 2X ResNet20 with W2A2 quantization still experiences reduced STDEV even if its weights are initialized in the same way the 1X ResNet model is initialized.

Finally, we confirm that the input activation's reduced dynamic range results in improved test accuracy. As shown in Fig.1c, 1X ResNet20 model suffers large accuracy degradation from quantization (-2.22%) while 2X ResNet20 model experiences relatively small accuracy degradation (-0.84%). Note that this result is consistent with WRPN (Mishra et al. (2018)). However, we argue that such increased robustness against quantization errors is due to the mechanism that the expanded channel structure reduces the activation's dynamic range.

From these experimental results, we conjecture that expanded channel structure plays a crucial role in compensating quantization errors. This finding suggests that if we employ channel expansion in the NAS framework, we can adapt the channels of the layers to find a new network structure that is more robust to uniform-precision quantization when trained from scratch. In the next section, we investigate a proper way of incorporating channel expansion.

## 4.2    IMPACT OF CHANNEL EXPANSION ON DISTINCT CHANNEL SEARCH

The previous section provides critical reasoning behind the success of channel expansion of QDNN like WRPN. However, it is not practical to expand the channels of all the layers, since it will quadratically increase the computational complexity. The main idea of NCE is to allow channel expansion only when the selection of more number of channels is desirable for compensating quantization errors; otherwise, the channels are pruned to meet the overall hardware constraints. In this section, we show that the channel expansion mechanism of NCE propels such compensation.

We first show that the channel selection preference can be observed by the gradient w.r.t. the search parameters. As explained in Sec. 3, the critical trade-off explored during the search is between the cross-entropy loss and the hardware constraint loss. In particular, a layer that is sensitive to the cross-entropy may select a large number of channels. In other words, the search parameters associated with a large number of channels, like $\alpha_7$ or $\alpha_8$, receive large negative gradients.[3] As an example, Fig. 2a shows the gradients of the search parameters during the TAS search of CIFAR10-ResNet20 in full-precision. Note that the search parameter associated with the maximum number of channels ($\alpha_8$) initially receives the negative gradients. In contrast, the search parameter with the least number of

---

[2]I.e., the weights are initialized with the normal distribution with standard deviation inversely proportional to the square root of the number of channels.

[3]I.e., $\alpha = \alpha - \eta * grad_\alpha$. Thus the negative gradients increase the magnitude of the search parameter.

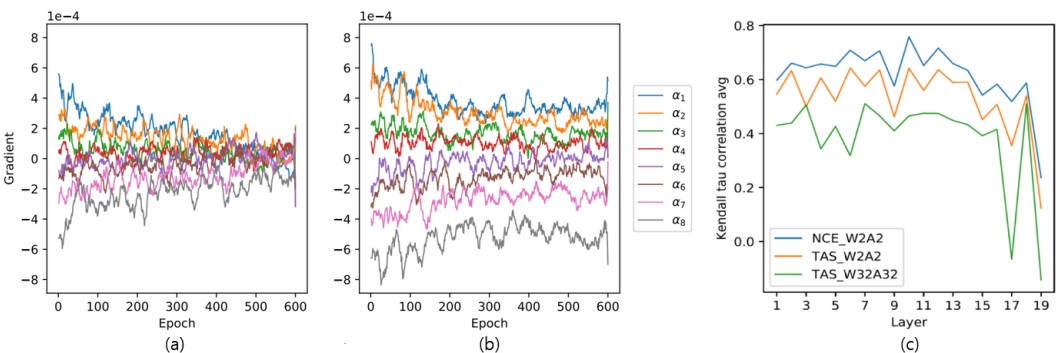

Figure 2: Experiments on CIFAR10-ResNet20 that shows gradients of cross-entropy loss w.r.t. search parameters ($\alpha_1 \sim \alpha_8$) of a layer during search: (a) in full-precision, (b) with 2-bit quantization. (c) Kendall rank-correlation score of all layers.

channels ($\alpha_1$) receives positive gradients. From this trend, we can conjecture that this layer initially prefers a large number of channels, but the preference diminishes over the epochs of search.

Next, we show that quantization during search excels the preference for a large number of channels. From the same experimental settings, if we apply quantization during the search, the preference to a large number of channels becomes more distinctive, as shown in Fig. 2b. This phenomenon is because the quantization errors affect the cross-entropy loss more. To quantify this channel preference, we calculate the Kendall rank correlation score for the gradients of the search parameters and average over the search epochs; the more consistent preference to a large number of channels, the higher the Kendall score. Fig. 2c shows the layer-wise Kendall score with and without quantization. Note that the Kendall score is increased if the quantization is applied during the architecture search. This increased Kendall score implies that quantization drives the search parameters toward a strong preference to a larger number of channels.

As motivated by this phenomenon, we augment TAS's search space to allow channel expansion. Thanks to this new search space, channel expansion can happen selectively for the layer with a strong preference for many channels. Interestingly, the new search space gains the higher Kendall score, as shown in Fig. 2c, indicating that the search space allows preference to an even larger number of channels. In other words, our simple yet novel search space of selectively expanding the channels can mitigate the limit in choosing a larger number of channels, opening up more significant opportunities for compensating quantization errors.

### 4.3 DISCUSSION ON NEURAL CHANNEL EXPANSION

So far, we explained how NCE was devised to facilitate the channel expansion for compensating the quantization error. In this section, we further investigate two aspects of NCE to understand its strengths better.

#### 4.3.1 BENEFIT OF QUANTIZATION-AWARE ARCHITECTURE SEARCH

We first show that NCE can reflect quantization during the search to find a better structure for quantization. As we discussed in the previous section, quantization affects the gradients w.r.t. the search parameters, resulting in the difference in the network structures after the search. In Fig.3a, we ran NCE for CIFAR10-ResNet20 with or without W2A2 quantization during the search. Then we took the models after each search and trained them from scratch with or without W2A2 quantization. After the full-precision training, both networks (searched with or without quantization) achieve the same accuracy level. In the case of W2A2, however, the network searched with quantization achieves noticeable gain in average accuracy over the network searched without quantization. Fig. 3b shows the difference in the channel section between the model searched with (=W2A2) and without quantization(=W32A32) where W2A2 prefers more channels in the later layers. This demonstrates that NCE can perform quantization-aware architecture search.

Figure 3: (a) Test accuracy after search with or without quantization. (b) Model structure after search with or without quantization. (c) Accuracy vs. FLOPs for different channel expansion strategies.

### 4.3.2 BENEFIT OF SELECTIVE CHANNEL EXPANSION

There are two options to search for channel expansion. One option is to start with the enlarged channels as the search space then prune, while the other option is to expand the channel selectively as NCE does. To understand the selective expansion's effectiveness, we constructed an experiment when the model is searched with 1) 1X channels accompanied by eight search parameters for each layer but with NCE, and 2) 2X channels accompanied by 16 search parameters layer. Note that the search space of the 1X-NCE case is strictly a subset of the 2X case. Therefore, if the 1X-NCE finds a suitable network structure, 2X should also find an equally good one. However, it turns out that the search results of the 2X case are inferior to NCE. As shown in Fig. 3c, for the same target FLOPs, the network structures found by the 2X case achieve inferior test accuracy than NCE. In Appendix, we confirm that the observed accuracy gain is indeed originated from the proposed selective increase in the search space.

## 5 EXPERIMENTS

### 5.1 EXPERIMENTAL SETTINGS

NCE is evaluated with popular CNNs trained on CIFAR10 and IamgeNet datasets. We employ PACT (Choi et al. (2018)) as the main quantization scheme. NCE is implemented in PyTorch based on the TAS framework[4]. For CIFAR10 experiments, we conducted 200 epochs of warm-up followed by NCE search for 600 epochs. we used threshold ($T$) of 0.3 and constraint coefficient of 2. For ImageNet experiments, we randomly choose 50 classes from the original 1000 classes to reduce the training time, similar to Wu et al. (2019). We conducted 40 epochs of warm-up followed by search for 110 epochs. we used threshold ($T$) of 0.19 and constraint coefficient of 1.5. We optimized the weight via SGD and the architecture parameters via Adam. Regularization coefficient of PACT is 0.001. After search, the candidate model is derived by the "winner-takes-all" strategy. Unless noted otherwise, all experiments on CIFAR10 are repeated for three times and the average test accuracy is reported. For fair comparison with prior work (e.g., Choi et al. (2018); Jung et al. (2019)), the first/last and short-cut layers are not considered for 2-bit uniform-precision quantization. More detailed information about the experimental settings can be found in Appendix.

### 5.2 CIFAR10 RESULTS

To evaluate NCE on CIFAR10, we employ four popular CNNs based on the ResNet structure and the other based on the VGG structure. For each network, we apply 2-bit quantization with and without NCE. As can be shown in Table 1, NCE consistently boosts the accuracy of QDNN w/o NCE by $0.35\% \sim 0.81\%$, demonstrating its benefit. Also, we demonstrate with CIFAR10-ResNet20 that the loss for hardware cost can be either FLOPs or the number of parameters (PARAM); Table 1 shows that ResNet20 could achieve higher accuracy when NCE searches with the PARAM loss. This flexibility in hardware loss can be convenient when targeting hardware platforms with specific needs.

---

[4]https://github.com/D-X-Y/AutoDL-Projects

| Network | W32A32 | w/o NCE | | | w/ NCE | | |
|---|---|---|---|---|---|---|---|
| | | W2A2 | FLOPs | PARAM | W2A2 | FLOPs | PARAM |
| ResNet20 | 92.88% | 90.82% | 40.81M | 0.27M | 91.63% | 39.94M | 0.42M |
| | | | | | 91.81% | 45.40M | 0.27M* |
| ResNet32 | 93.81% | 92.22% | 69.12M | 0.47M | 92.71% | 64.26M | 0.57M |
| ResNet56 | 94.26% | 93.08% | 125.75M | 0.86M | 93.43% | 123.04M | 0.74M |
| VGG16 | 94.24% | 93.48% | 313.2M | 14.72M | 93.94% | 302.96M | 5.01M |

Table 1: CIFAR10: Comparision on test accuracy and computational complexity for 2-bit QDNN with or without NCE (*: constrained with PARAM loss; otherwise constrained with FLOPs).

| Network | Method | Top-1 Acc | Top-5 Acc | FLOPs | PARAM |
|---|---|---|---|---|---|
| ResNet18 | *Full precision* | *70.56%* | *89.88%* | 1.814G | 11.69M |
| | w/o NCE(Ours) | 64.08% | 86.47% | 1.814G | 11.69M |
| | **w/ NCE(Ours)** | **66.17%** | **86.75%** | **1.747G** | **12.57M** |
| | LSQ | 67.6% | 87.6% | | |
| | QIL | 65.7% | - | | |
| | LQ-Nets | 64.9% | 85.9% | 1.814G | 11.69M |
| | PACT | 64.4% | 85.6% | | |
| | EdMIPS | 65.9% | 86.5% | | |
| ResNet50 | *Full precision* | *76.82%* | *93.33%* | 4.089G | 25.56M |
| | w/o NCE(Ours) | 72.36% | 90.81% | 4.089G | 25.56M |
| | **w/ NCE(Ours)** | **74.03%** | **91.63%** | **3.932G** | **17.66M** |
| | LSQ | 73.7% | 91.5% | | |
| | LQ-Nets | 71.5% | 90.3% | 4.089G | 25.56M |
| | PACT | 72.2% | 90.5% | | |
| | EdMIPS | 72.1% | 90.6% | | |

Table 2: ImageNet: Top-1/Top-5 accuracy and computational complexity for 2-bit (W2A2) QDNN with or without NCE compared with the other state-of-the-art uniform (LSQ, QIL, LQ-Nets, PACT) and mixed-precision (EdMIPS) quantization methods.

## 5.3 IMAGENET RESULTS

We further evaluate NCE with ResNet structures on ImageNet by comparing it with state-of-the-art QDNN methods. We take reported Top-1 and Top-5 accuracy from the state-of-the-art uniform-precision quantization techniques; LSQ(Esser et al. (2020)), QIL(Jung et al. (2019)), LQ-Nets(Zhang et al. (2018)), and PACT(Choi et al. (2018)). We also include recent work on mixed-precision quantization; EdMIPS(Cai & Vasconcelos (2020)). The comparison of accuracy is summarized in Table 2. Note that NCE consistently improves the accuracy of 2-bit quantized models (=w/o NCE) by large margins (1.67% ∼ 2.09%) thanks to the structure's adaptation. Although the PACT quantizer's performance in NCE is inferior to the more recent techniques (e.g., LSQ and QIL), NCE could boost the accuracy to the extremely competitive level. In the case of ResNet50, NCE even surpasses the best-to-date QDNN method, LSQ, in terms of accuracy, FLOPs, and the number of parameters involved in the network. This superior performance indicates that NCE is a promising solution for compensating the accuracy degradation of QDNN.

## 6 CONCLUSION

In this work, we propose a novel approach that explores the neural network structure to achieve robust inference accuracy while using the simple uni-precision arithmetic units. The key idea is to explore the network structure under the impact of uni-precision quantization. Our novel differentiable neural architecture search employs the search space that can shrink and expand the channels so that the more sensitive layers can be equipped with more channels. At the same time, the overall resource requirements (e.g., FLOPs) are maintained. We provide substantial insights and empirical supports that the proposed method can achieve superior robustness in ultra-low uni-precision quantization for networks in CIFAR10 and ImageNet. In particular, we demonstrate a superior ResNet50 architecture that achieves higher accuracy for 2-bit inference with smaller FLOPs and the parameter size.

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

| Max Num Channels | Accuracy | FLOPs | PARAM |
|---|---|---|---|
| 1.25X ResNet20 | 91.45% | 40.19M | 0.28M |
| 1.50X ResNet20 | 91.82% | 39.63M | 0.34M |
| 2.00X ResNet20 | 91.63% | 39.94M | 0.42M |

Table 3: Impact of constraining the maximum number of channels for NCE search.

Ritchie Zhao, Yuwei Hu, Jordan Dotzel, Chris De Sa, and Zhiru Zhang. Improving neural network quantization without retraining using outlier channel splitting. In *International Conference on Machine Learning*, pp. 7543–7552, 2019.

Shuchang Zhou, Yuxin Wu, Zekun Ni, Xinyu Zhou, He Wen, and Yuheng Zou. Dorefa-net: Training low bitwidth convolutional neural networks with low bitwidth gradients. *arXiv preprint arXiv:1606.06160*, 2016.

Yiren Zhou, Seyed-Mohsen Moosavi-Dezfooli, Ngai-Man Cheung, and Pascal Frossard. Adaptive quantization for deep neural network. *arXiv preprint arXiv:1712.01048*, 2017.

Barret Zoph and Quoc V Le. Neural architecture search with reinforcement learning. *arXiv preprint arXiv:1611.01578*, 2016.

Barret Zoph, Vijay Vasudevan, Jonathon Shlens, and Quoc V Le. Learning transferable architectures for scalable image recognition. In *Proceedings of the IEEE conference on computer vision and pattern recognition*, pp. 8697–8710, 2018.

# 7 APPENDIX

## 7.1 EXPERIMENTAL DETAILS

In this section, we summarize the details of our training settings. The CIFAR10 dataset (Krizhevsky Hinton (2010)) is an image classification benchmark containing 32x32 pixel RGB images. It consists of 50K training and 10K test image sets. We used RandomHorizontalFlip and Normalization for preprocessing. We used stochastic gradient descent (SGD) with a momentum of 0.9 and learning rate starting from 0.1 and scheduled by cosine annealing to update network weight. We used Adam optimizer and learning rate of 0.001 to update the search parameter. L2-regularizer with the decay of 0.0004 is applied to weight, and 0.001 is applied to the search parameter. The mini-batch size of 256 is used.

The ImageNet dataset (Russakovsky et al. (2015)) consists of 1000-categories of objects with over 1.2M training and 50K validation images. We used RandomResizedCrop, RandomHorizontalFlip and Normalization for preprocessing. We used stochastic gradient descent (SGD) with a momentum of 0.9 and learning rate starting from 0.1 and scaled by 0.1 at epoch 30, 60, 85, 95 to update network weight. We used Adam and a learning rate of 0.001 to update the search parameter. L2-regularizer with the decay of 0.0001 is applied to weight, and 0.001 is applied to the search parameter. The mini-batch size of 256 is used.

## 7.2 ABLATION STUDY

In this section, we present various ablation study and analysis to better understand NCE.

### 7.2.1 IMPACT OF CONSTRAINING MAXIMUM NUMBER OF CHANNELS FOR NCE SEARCH

In Sec. 5, we evaluated NCE with the maximum number of channels as 2X of the original network size. In this setting, NCE sometimes found the networks with the hardware cost higher than the original network if it is not constrained[5], especially when the network size is relatively small; e.g., NCE for ResNet20-CIFAR10 achieved higher accuracy (=91.63%) when it is constrained with FLOPs (=40.81M), but with increased PARAM=0.42M.

---

[5]We followed the convention of restricting only one type of hardware cost, either PARAM or FLOPs.

| ResNet20 CIFAR10 W2A2 | Accuracy | PARAM | FLOPs |
|---|---|---|---|
| Full-Prec Baseline | 92.88% | 0.27M | 40.81M |
| W2A2 Baseline | 90.82% | 0.27M | 40.81M |
| W2A2 2X TAS (#search param=8) | 90.99% | 0.46M | 43.11M |
| W2A2 2X TAS (#search param=16) | 91.12% | 0.42M | 40.36M |
| W2A2 NCE (#search param=8∼16) | 91.63% | 0.42M | 39.94M |

Table 4: Comparison of quantization performance between NCE and 2X TAS with controlled search space size.

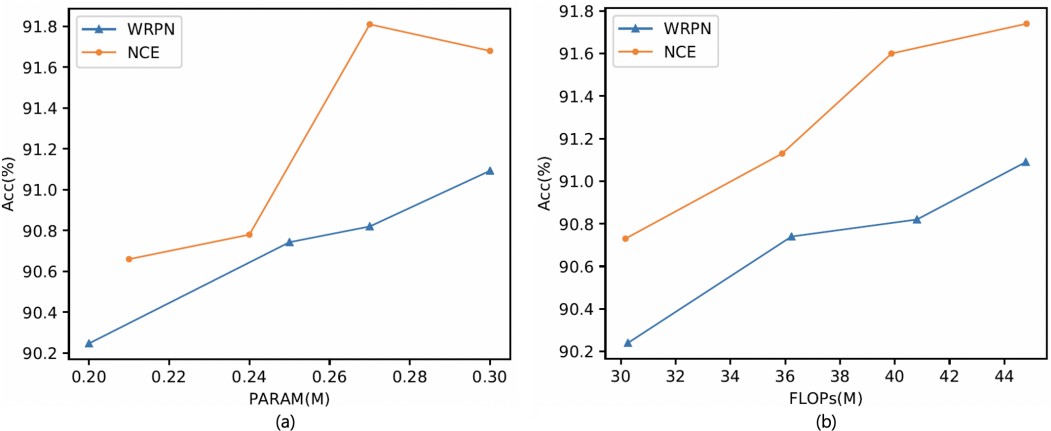

Figure 4: Comparison of quantization performance between NCE and WRPN with respect to (a) PARAM and (b) FLOPs on ResNet20-CIFAR10.

In this section, we extend our study to constrain both PARAM and FLOPs by fixing the maximum number of channels, limiting channel expansion. A detailed investigation about the impact of the maximum number of channels for the NCE search can is shown in Table 3. As the maximum number of channels decreases from 2X to 1.25X, PARAM also decreases while maintaining the overall accuracy.

### 7.2.2 COMPARISON OF QUANTIZATION PERFORMANCE BETWEEN NCE AND TAS

In Sec. 4.3.2, we discussed the benefits of the selective channel expansion over the prior channel pruning approach, TAS (Dong & Yang (2019)). We further investigate this issue to confirm that the observed accuracy gain originates from the selective channel expansion, not from the search space size. Table 4 shows the experimental results of 2-bit quantization on ResNet20-CIFAR10 with the number of channels adapted by either 2X TAS or NCE (with FLOPs=40M as the hardware constraint). Note that 2X TAS performs pruning from the 2X uniform channel expansion; thus, NCE's search space is a subset of 2X TAS's search space. The search space with both 8 and 16 search parameters per layer is considered for the controlled experiments. We can observe from the table that 2X TAS found the network with increased channels, achieving only modest accuracy gain. In contrast, NCE achieved a noticeable accuracy gain with slightly lower PARAM and FLOPs than 2X TAS. This experimental result demonstrates the benefit of selective channel expansion.

### 7.2.3 COMPARISON OF QUANTIZATION PERFORMANCE BETWEEN NCE AND WRPN

A straightforward method to explore the trade-off between the number of channels and the robustness of quantization error is to uniformly increase the number of channels for all the layers, as discussed in WRPN (Mishra et al. (2018)). We argue that NCE can explore a better accuracy vs. hardware cost trade-off by expanding the number of channels only for the necessary layers. To demonstrate it, we compare the trade-off curves between accuracy and the two popular hardware costs, PARAM and

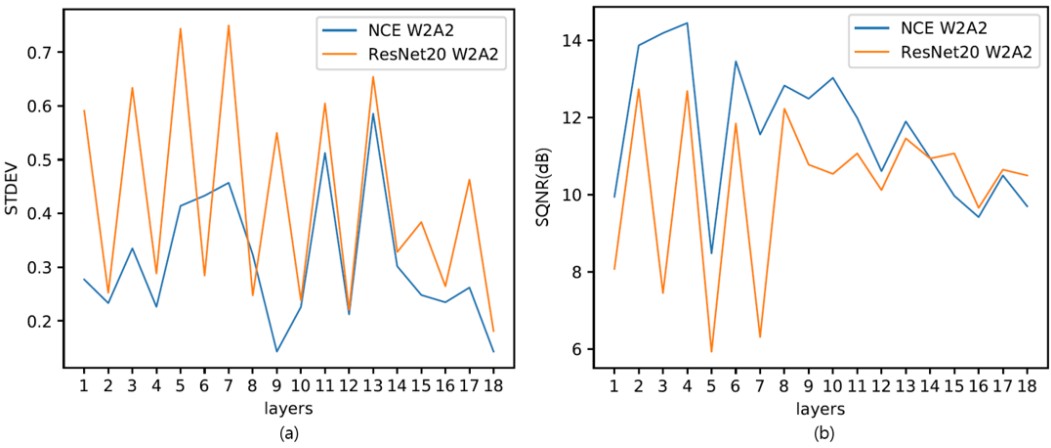

Figure 5: Comparison of STDEV and SQNR of activation between ResNet20-CIFAR10 and NCE when the models are quantized to 2-bit. (W2A2 Accuracy: ResNet20 (90.82%), NCE (91.63%))

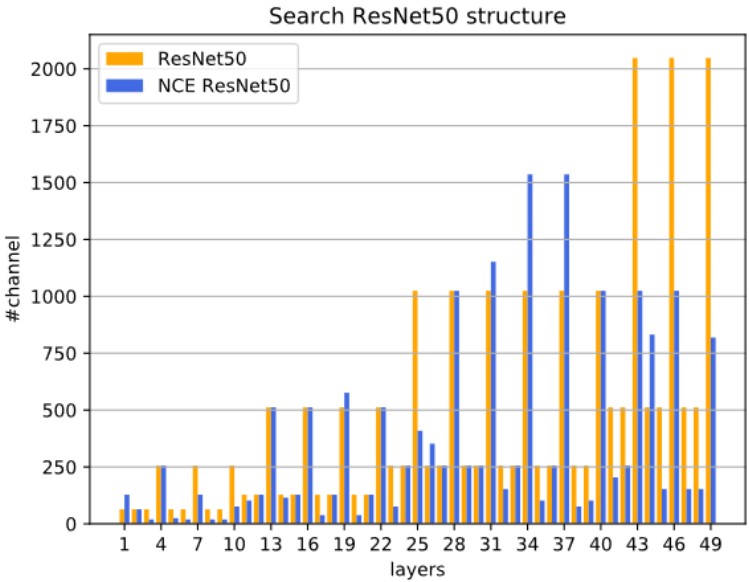

Figure 6: Comparison of structure of ResNet50-ImageNet before and after adaptation by NCE.

FLOPs. As shown in Fig. 4, NCE always achieves higher accuracy in terms of both PARAM and FLOPs, highlighting its superiority over WRPN.

### 7.2.4 RELATIONSHIP BETWEEN ACTIVATION DYNAMIC RANGE AND QUANTIZATION ERROR

In Sec. 4.1, we discussed the impact of quantization on the dynamic range of activation. To provide a concrete connection between the dynamic range of activation and the quantization error, we compare the signal-to-quantization-ratio (SQNR) measured when W2A2 is applied to activation of the ResNet20-CIFAR10 network and the one adapted by NCE. Note that SQNR is a well-known metric for the impact of quantization error on the model performance (Cai & Vasconcelos (2020)). Fig. 5 shows that NCE reduces the dynamic range of activation and increases the SQNR when the

| Threshold $T$ | Accuracy |
|:---:|:---:|
| 0.30 | 91.60% |
| 0.25 | 91.44% |
| 0.20 | 91.40% |
| 0.15 | 90.77% |

Table 5: The accuracy of the model NCE adapted from ResNet20-CIFAR10 with different channel expansion threshold, $T$.

| ResNet20-CIFAR10 | | Accuracy | FLOPs |
|:---:|:---:|:---:|:---:|
| Original structure (w/o NCE) | W32A32 | 92.88% | 40.81M |
| | W3A3 | 92.45% | |
| | W4A4 | 92.69% | |
| NCE | W3A3 | 92.66% | 39.07M |
| | W4A4 | 92.75% | 37.07M |

Table 6: 3-bit and 4-bit Quantization of ResNet20-CIFAR10 with and without NCE.

model is quantized, confirming its beneficial influence for compensating the quantization error and regain the model accuracy.

### 7.2.5 COMPARISON OF NETWORK STRUCTURE BEFORE AND AFTER ADAPTATION BY NCE

NCE adapts the channels of the original neural network to improve its robustness to the quantization error. As a result, the neural network searched by NCE has the number of channels expanded or reduced across the layers while maintaining overall shape. As an example, Fig. 6 shows the comparison of the number of channels of ResNet50-ImageNet before and after adaptation by NCE. Note that the original ResNet50 structure contains an abundant number of channels at the later layers, but these numbers of channels are artificially scaled as inversely proportional to the feature-map width. NCE successfully balances the number of channels to improve quantization accuracy while maintaining hardware constraints.

### 7.2.6 CHANNEL EXPANSION THRESHOLD

The channel expansion threshold $T$ in Algorithm 1 is the hyper-parameter that determines which layer to request the channel expansion. Given that there are 8 search parameters initialized uniformly, $T$ needs to be higher than $1/8 = 0.125$. Although it plays a critical role in enabling the selective expansion of the channels, the overall performance is not very sensitive to its choice. As shown in Table 5, there is a plateau in the model accuracy for the selection of $T > 0.15$. In practice, setting $T$ between 0.20 and 0.30 is sufficient.

### 7.2.7 QUANTIZATION WITH HIGHER BIT-PRECISION FOR NCE

To further understand the impact of NCE for higher bit-precision quantization, we experimented on ResNet20-CIFAR10. As shown in Table 6, NCE consistently finds neural networks with higher accuracy when they are trained with the same setting as the original network. Interestingly, FLOPs decreases as the bit-precision increases; this indicates that NCE is capable of effectively exploring the trade-offs between FLOPs and accuracy.

