# OpenReview forum: "Uniform-Precision Neural Network Quantization via Neural Channel Expansion"
_ICLR.cc/2021/Conference — Reject_

### Official Review · AnonReviewer1 · 2020-10-28
**Official Blind Review #1**

**Rating:** 5
**Confidence:** 5

**Review:**

In this paper, the authors propose neural channel expansion (NCE) to adjust the network structure to compensate for the performance degradation from uniform-precision quantization. Given a hardware constraint, the proposed NCE selectively expands the width for the quantization sensitive layers. Experiments on CIFAR-10 and ImageNet shows the effectiveness of the proposed NCE. However, the novelty of this paper is limited since the proposed method is just an extension of TAS. Besides, the comparisons between NCE and the existing methods are missing. My detailed comments are as follows.

Positive points:
1. Based on TAS, the authors propose a neural channel expansion (NCE) method to search for the width across the layers under uniform-precision quantization.

2. The authors provide an analysis of the proposed NCE on the effect of channel expansion for compensation of quantization errors.

3. Experimental results on CIFAR-10 and ImageNet demonstrate that NCE is able to improve the performance of the quantized networks.

Negative points:
1. The novelty of this paper is limited. The authors only make minor modifications to TAS [1] and apply it to network quantization. However, neither of the modification on TAS or the application of channel expansion on network quantization is significant.

2. This paper is quite similar to the work of Shen et al. [2], which also explores the optimal layer width for quantization. Thus, it would be better to compare the proposed method with Shen et al. ’s.

3. As mentioned in Section 2, there are several existing works [3][4][5] that reducing the quantization error by increasing the widths of the network. More comparison between the proposed method and the existing works would make this paper more convincing.

4. In Section 4.1, the authors claim that “The substantial increase of STDEV for W2A2 implies that large quantization errors would occur when input activation is quantized”. However, Figure 1a only provides the results of W2A2 and W32A32. Thus, it cannot illustrate whether the large quantization error results from the quantization of activation or not. At least, the results of W32A2 and W2A32 should be provided.

5. In Section 4.1, the authors argue that improved performance is resulted from the reduction of input activation’s dynamic range according to Figures 1a and 1c. However, the STDEV concerning “2X ResNet20 (W32A32)” is missing in Figure 1a. To compare the dynamic range reduction of ResNet20 with 1x and 2x width, more results regarding the STDEV of “2x ResNet W32A32” would make this paper more convincing.

6. As discussed in Section 4.1, the improved performance comes from the reduction of activation’s dynamic range. However, it is not clear how the dynamic range of activation impacts model performance. It is better to give more theoretical analyses and explanations regarding the results.

7. The authors only apply NCE to 2-bit quantization. It will be better if the authors would provide results on more bitwidth quantization, e.g., 4-bit or 1-bit.

8. In Algorithm 1, the threshold T is important to the proposed method. According to the experimental settings on CIFAR-10 and ImageNet, the chosen of threshold T is tricky. More results on different values of T are required.

9. In the first line of Algorithm 1, the n(D_val) should be n(D_arch).

References

[1] Xuanyi Dong et al. Network pruning via transformable architecture search. NeurIPS 2019.

[2] Mingzhu Shen, Kai Han, et al. Searching for accurate binary neural architectures. ICCV Workshop 2019.

[3] Asit Mishra et al. Wrpn: wide reduced-precision networks. ICLR 2018.

[4] Ritchie Zhao et al. Improving neural network quantization without retraining using outlier channel splitting. ICML 2019.

[5] Hanmin Park et al. Cell division: weight bit-width reduction technique for convolutional neural network hardware accelerators. ASPDAC 2019.

---

> ### Author Response · Authors · 2020-11-24
> **Answers to Reviewer1's comments**
>
> We thank the reviewer for his/her constructive and insightful comments. Here we prepared the answers to all the questions raised.
>
>
> Q) Concerns about novelty
>
> A) We concur with the reviewer's concern that our contributions are not sufficiently differentiated. In this regard, we would like to argue about the novelty of our paper in threefolds.
>
> - NCE is NOT a typical method that simply exploits the increased robustness from the channel expansion. With substantial empirical evidence, we claim that the proposed method finds a non-trivial network structure that conventional approaches like increasing channels uniformly (WRPN) or pruning from the enlarged backbone structure (TAS) cannot find. As concrete evidence, Fig. 3c and 4 show that NCE achieves superior trade-offs between the accuracy and the hardware cost (in terms of both PARAM and FLOPs) compared to TAS and WRPN. We further augmented the controlled study of Fig. 3c in Table 4, which demonstrates the superiority of NCE over TAS in channel search.
>
> - The search space of NCE is NOT a straightforward extension of the channel pruning of TAS. Instead, the proposed search space (called neural channel expansion) is motivated by an insightful observation that the structure (not the neural network's parameter values) directly impacts the dynamic range of activation, resulting in reduced quantization error. As concrete evidence, we added Fig. 5 to show that NCE can increase SQNR, which leads to improved model accuracy.
>
> - NCE achieves significant accuracy gain with reduced hardware cost for 2-bit quantization, especially with the model of sufficient size (e.g., VGG16-CIFAR10, ResNet50-ImageNet). We want to emphasize that NCE even outperforms the SOTA mixed-precision model (EdMIPS, CVPR2020), as shown in Table 2.
>
>
> Q) This paper is quite similar to the work of Shen et al. [2], which also explores the optimal layer width for quantization
>
> A) Thanks for letting us know about relevant work. Here is a little detail on the performance comparison between NCE and Shen et al.
> - Both methods use high-precision for the first/last layers
> - [Shen et al.] employ ops-scaling by 64 for their binary ops (ops-scaling: 64-ops in binary computation accounts for one FLOP).
> - Thus, NCE takes ops-scaling conservatively by 16 for our 2-bit ops.
> - [Shen et al.]'s baseline (full-prec) performance: 93.48% with 608M FLOPs.
> - [Shen et al.]'s VGG-Auto-B performance: 93.06% (-0.42% degradation) with 59.3M FLOPs.
> - NCE's baseline (full-prec) performance: 94.24% with 313.2M FLOPs
> - NCE's W2A2 performance: 93.94% (-0.30%) with 20.6M FLOPs
>
> Therefore, NCE achieves slightly lower accuracy degradation (and higher absolute accuracy) with 3X smaller FLOPs than [Shen et al.].
>
>
> Q) More comparison between the proposed method and the existing works would make this paper more convincing
>
> A) Thanks for the constructive suggestion. We now provide a performance comparison with WRPN, one of the most well-known prior works exploiting channel expansion for improved quantization accuracy. As shown in Fig. 4, NCE outperforms WRPN in terms of both PARAM and FLOPs.
>
> Q) Figure 1a only provides the results of W2A2 and W32A32. Thus, it cannot illustrate whether the large quantization error results from the quantization of activation or not. At least, the results of W32A2 and W2A32 should be provided. ... To compare the dynamic range reduction of ResNet20 with 1x and 2x width, more results regarding the STDEV of “2x ResNet W32A32” would make this paper more convincing. ... However, it is not clear how the dynamic range of activation impacts model performance. It is better to give more theoretical analyses and explanations regarding the results.
>
> A) Thanks for another constructive suggestion. We now provide separate plots to show 1) the impact of quantization on the dynamic range of activation, and 2) the impact of channel expansion on the dynamic range of activation. It is clearly shown in Fig. 1a-b that especially activation quantization significantly increases the dynamic range, which can be effectively compensated by the channel expansion. We further provide in Fig. 5 the link between the increased dynamic range of activation and the quantization error in terms of SQNR.
>
>
> Q) The authors only apply NCE to 2-bit quantization. It will be better if the authors would provide results on more bitwidth quantization, e.g., 4-bit or 1-bit.
>
> A) Thanks for the suggestion. We now include 3- and 4-bit quantization results for NCE on ResNet20-CIFAR10 in Table 6. Note that the quantization with NCE outperforms the case without NCE. Interestingly, NCE could reduce FLOPs more as the bit-precision increases, indicating that NCE is capable of effectively exploring the trade-offs between FLOPs and accuracy.

---

> > ### Author Response · Authors · 2020-11-24
> > **Answers to Reviewer1's comments (Cont'd)**
> >
> > Q) the threshold T is important to the proposed method. According to the experimental settings on CIFAR-10 and ImageNet, the chosen of threshold T is tricky. More results on different values of T are required.
> >
> > A) Thanks a lot for the suggestion. We now provide the ablation study for T in Table 5. Note that the performance of NCE is not very sensitive to it; In practice, setting T between 0.20 and 0.30 is sufficient.
> >
> >
> > Q) Typos and errors
> >
> > A) Thanks for pointing them out. We will further proofread the text carefully to remove all the errors.

---

### Official Review · AnonReviewer3 · 2020-10-29
**Review for Uniform-Precision Neural Network Quantization via Neural Channel Expansion**

**Rating:** 6
**Confidence:** 5

**Review:**

### Overview
In this paper, the authors proposed selectively channel pruning and expansion via neural architecture search to make an off-the-shelf neural network more robust to quantization. Under 2-bit quantization, the proposed method outperforms the original model at similar or smaller FLOPs and model size.

### Clarity
Overall, the paper shows the concept clearly. The method itself is not complicated. But some definitions are missing to make the paper self-contained. For example, the implementation of channel-wise interpolation (CWI) is not provided, and not referred to existing literature. The presentation of Figure 2(a)(b) makes it hard to see the trend. It would be better to plot the temporally averaged curve (potentially with std) for better visualization.

### Pros
1. The paper is generally well written and clearly expressed.

2. It focuses on an important topic: neural architecture search under (low-bit) model quantization, which has been less explored before. The paper shows that considering the quantization during the search can improve the final accuracy.

3. The proposed method shows good results. On ImageNet, the optimized model outperforms the original version at similar or lower cost after quantization.

### Cons
1. My first concern is the novelty. As acknowledged by the authors, improving quantization with channel expansion has been widely studied [Mishara et al. 2018, Zhao et al. 2019, Park&Choi 2019] and proves to be effective. The authors claimed that "none of the approaches shed light on understanding the benefit of the increased number of channels when a network is trained for quantization", which I think is an overclaim. The reduction of dynamic range from increased channels is already well demonstrated in previous work (e.g., Zhao et al. 2019); while the authors verify the conclusion again in this work. The author did prove that the modified/expanded architecture works well when trained from scratch, but it is not a very surprising finding since networks with higher capacity are more robust to quantization.
As for the NCE method itself, it is like an extension of TAS by allowing expansion in search space and under quantization. The core novelty seems to the extended search space with channel expansion, but the other parts are largely the same.

2. A trivial alternative would be to apply TAS using the enlarged search space (Sec 4.3.2) with theoretically the same optimal solution. The ablation study shows that the larger search space might hinder successful optimization, leading to worse results. I think using 16 search parameters makes the optimization too difficult; a fair comparison would be to still use 8 search parameters, with channel configurations using larger strides to compare the accuracy, so that we can exclude the factor of architecture search difficulty and focus on the search space design (which is the main contribution of the paper).

3. Regarding the experimental results, it outperforms the original model quantization using various methods. However, the authors did not compare to existing results that modify/expand the network architecture for better quantization accuracy (e.g., Zhao et al. 2019). Without comparing to existing methods on the same topic, it would be difficult to evaluate the contribution.

4. The author mentioned that the expanded architecture itself is beneficial for quantization. For results in Table 2, if we train the optimized architecture from scratch, will we still have the same advantage?

5. Minor question: it is unclear how the batch normalization layers are processed. Are they fused into convolutions or kept as float-point?


I hope to see the authors' feedback for the final evaluation. Thanks!

EDIT: Raised score from 4 to 6 after the reading authors' response. The response clarifies some of the novelty issues, and it clearly shows the advantage compared to previous methods like TAS. However, I still have concerns about the novelty; the insight why the proposed method is better than TAS is still not very clear to me. I hope the author can further improve the draft for the final version.

---

> ### Author Response · Authors · 2020-11-24
> **Answers to Reviewer3's comments**
>
> We thank the reviewer for his/her constructive and insightful comments. Here we prepared the answers to all the questions raised.
>
> Q) Concerns about novelty
>
> A) We concur with the reviewer's concern that our contributions are not sufficiently differentiated. In this regard, we would like to argue about the novelty of our paper in threefolds.
>
> - NCE is NOT a typical method that simply exploits the increased robustness from the channel expansion. With substantial empirical evidence, we claim that the proposed method finds a non-trivial network structure that conventional approaches like increasing channels uniformly (WRPN) or pruning from the enlarged backbone structure (TAS) cannot find. As concrete evidence, Fig. 3c and 4 show that NCE achieves superior trade-offs between the accuracy and the hardware cost (in terms of both PARAM and FLOPs) compared to TAS and WRPN. We further augmented the controlled study of Fig. 3c in Table 4, which demonstrates the superiority of NCE over TAS in channel search.
>
> - The search space of NCE is NOT a straightforward extension of the channel pruning of TAS. Instead, the proposed search space (called neural channel expansion) is motivated by an insightful observation that the structure (not the neural network's parameter values) directly impacts the dynamic range of activation, resulting in reduced quantization error. As concrete evidence, we added Fig. 5 to show that NCE can increase SQNR, which leads to improved model accuracy.
>
> - NCE achieves significant accuracy gain with reduced hardware cost for 2-bit quantization, especially with the model of sufficient size (e.g., VGG16-CIFAR10, ResNet50-ImageNet). We want to emphasize that NCE even outperforms the SOTA mixed-precision model (EdMIPS, CVPR2020), as shown in Table 2.
>
>
> Q) The authors claimed that "none of the approaches shed light on understanding the benefit of the increased number of channels when a network is trained for quantization", which I think is an overclaim.
>
> A) Thanks for pointing out the ambiguity of our claim. The point we would like to make was that our discovery about the link between the dynamic range of activation and the network structure provides a new perspective to the improved robustness of the channel expanded models to the quantization errors. Note that Zhao et al. (2019) and Park & Choi (2019) are focusing on improving post-training quantization with channel splitting. Thus it is not straightforward to extend their work for quantization-aware training, which is necessary for ultra-low bit DNN inference. We revised the entire paragraph to clarify this point.
>
>
> Q) The reduction of dynamic range from increased channels is already well demonstrated in previous work (e.g., Zhao et al. 2019); while the authors verify the conclusion again in this work.
>
> A) Note that the context of [Zhao et al. 2019, Park&Choi 2019] is different from ours; they are focusing on post-training quantization, numerically manipulating the channels for a given pre-trained model. Thus, these approaches may not be applicable for quantization-aware training (QAT), a necessary procedure for ultra-low precision quantization of neural nets. We empirically showed that in the context of QAT, manipulation of weight parameters (initialized with a dynamic range based on 1X or 2X channels in Fig. 1b) does not affect the dynamic range of activation when the model is quantized to 2-bit.
>
>
> Q) The author did prove that the modified/expanded architecture works well when trained from scratch, but it is not a very surprising finding since networks with higher capacity are more robust to quantization.
>
> A) We agree with the reviewer that the increased capacity via channel expansion was the previous belief about the improved robustness to the quantization. However, there was no prior work connecting this robustness to reducing the dynamic range of activation and thus quantization error. (In Fig. 5, we showed that NCE not only reduces the dynamic range of activation (in terms of STDEV) but also increases the SQNR.) Although we did not provide theoretical proof, we demonstrated that such observation is global. This more explicit demonstration of the relationship between channel structure and the quantization error motivated us to seek the proper channel expansion search for robust yet computationally economic neural networks.

---

> > ### Author Response · Authors · 2020-11-24
> > **Answers to Reviewer3's comments (Cont'd)**
> >
> > Q) A trivial alternative would be to apply TAS using the enlarged search space (Sec 4.3.2) with theoretically the same optimal solution. The ablation study shows that the larger search space might hinder successful optimization, leading to worse results. I think using 16 search parameters makes the optimization too difficult; a fair comparison would be to still use 8 search parameters, with channel configurations using larger strides to compare the accuracy, so that we can exclude the factor of architecture search difficulty and focus on the search space design
> >
> > A) Thanks for letting us clarify our comparison of NCE with TAS. As you pointed out, the search space size was different in the experiment of Fig. 3c. We now provide a more controlled experiment (as suggested) in Table 4, where NCE always outperforms TAS approaches regardless of the number of search parameters. This result supports our claim that NCE's channel expansion mechanism allows superior exploration of the channel search space.
> >
> >
> > Q) Without comparing to existing methods on the same topic, it would be difficult to evaluate the contribution.
> >
> > A) Thanks for the constructive suggestion. We now provide a performance comparison with WRPN, one of the most well-known prior works exploiting channel expansion for improved quantization accuracy. As shown in Fig. 4, NCE outperforms WRPN in terms of both PARAM and FLOPs.
> >
> >
> > Q) The author mentioned that the expanded architecture itself is beneficial for quantization. For results in Table 2, if we train the optimized architecture from scratch, will we still have the same advantage?
> >
> > A) Thanks for letting us clarify the experimental settings. All the experimental results we conducted in this paper (including Table 2) re-train the model searched via NCE from scratch; the hyper-parameters are mostly the same as the baseline (more details in Sec. 5.1 and Sec. 7.1). In fact, we observed the same phenomena (discussed in Sec 4) in all the experiments, including ResNet50-ImageNet.
> >
> > Q) Minor question: it is unclear how the batch normalization layers are processed. Are they fused into convolutions or kept as float-point?
> >
> > A) In this work, we follow the convention of prior QAT methods (PACT, QIL, LSQ, ...) to compute batchnorm in high-precision separately.
> >
> >
> > Q) Clarity issues: Figure 2(a)(b), missing to make the paper self-contained, etc.
> >
> > A) Thanks for pointing them out. We updated Fig. 2 to improve clarity; we will further proofread the text carefully to remove all the missing terms and definitions.

---

### Official Review · AnonReviewer4 · 2020-10-30
**Good idea, but simple. Results comparison has flaws.**

**Rating:** 6
**Confidence:** 4

**Review:**

The authors propose neural channel expansion (NCE), a neural architecture search (NAS) and quantization method. Existing NAS+Q methods typically search for the architecture of the DNN along with the precision at each layer, maximizing accuracy while respecting some kind of hardware constraint. The result is a DNN with mixed-precision, which is challenging for most existing hardware (which only support one or a few precisions). NCE keeps precision the same in each layer, and instead uses the precision sensitivity signal in the NAS to adjust the width of the layer (expand or shrink). The result is uniform-precision, hardware-friendly DNN.

NCE works by first training normally (with quantization) for 40 warmup epochs, followed by 110 epochs of search. At the end of each search epoch, NCE adjusts the channels search parameter in each layer. A few experiments show convincingly that a wider layer is indeed less sensitive to quantization. Thus using the sensitivity-to-quantization signal to adjust layer width is a good idea.

Experiments on CIFAR-10 show NCE can boost quantized accuracy at 2w2a by up to 0.8%, and in the case of VGG16 trim unnecessary params. On ImageNet there is also some accuracy improvement, though only a little bit over LSQ. And for ResNet-50 NCE again can reduce param size.

The paper is that the idea is fairly simple, and the results are not too impressive. LSQ seems to already do very well and it also uses uniform quantization. One major issue I have with the comparison in Tables 1 and 2 is that, on some of the smaller networks (ResNet-32 for CIFAR and ResNet-18 for ImageNet) the NCE result has more params than the other methods. This is potentially unfair as a larger network is almost always more accurate. I think you should uniformly increase channel widths in at least the "w/o NCE" baseline to see if the accuracy boost is from NCE learning the layer sensitivities or just from a bigger model.

The results on larger networks (VGG and ResNet-50) is much more compelling, showing that NCE can trim unnecessary params while improving accuracy. More results like this would make the paper more convincing. I would also like to see exactly which layers were reduced in size on these networks.

Another issue is that despite being a NAS work, there aren't NAS baselines in the comparison. I understand that NCE only requires one training run while the original NAS required many retrainings. But I believe HAQ (Wang et al 2019) and DARTS (Liu et al 2018) are both NAS techniques for mixed-precision quantization that require only one training run. The authors should include a comparison against such methods or discuss why it isn't needed.

Minor issues:
 - Section 4.3.2 Typo: "results of the 2X case are inferior to the 2X case"
 - Table 2, ResNet-18, you highlighted your own result but LSQ seems to be better in accuracy and param size?

EDIT: Raised score from 4 to 6 after the authors clarified some points and added additional experiments.

---

> ### Author Response · Authors · 2020-11-24
> **Answers to Reviewer4's comments**
>
> We thank the reviewer for his/her constructive and insightful comments. Here we prepared the answers to all the questions raised.
>
> Q) Concerns about novelty
>
> A) We concur with the reviewer's concern that our contributions are not sufficiently differentiated. In this regard, we would like to argue about the novelty of our paper in threefolds.
>
> - NCE is NOT a typical method that simply exploits the increased robustness from the channel expansion. With substantial empirical evidence, we claim that the proposed method finds a non-trivial network structure that conventional approaches like increasing channels uniformly (WRPN) or pruning from the enlarged backbone structure (TAS) cannot find. As concrete evidence, Fig. 3c and 4 show that NCE achieves superior trade-offs between the accuracy and the hardware cost (in terms of both PARAM and FLOPs) compared to TAS and WRPN. We further augmented the controlled study of Fig. 3c in Table 4, which demonstrates the superiority of NCE over TAS in channel search.
>
> - The search space of NCE is NOT a straightforward extension of the channel pruning of TAS. Instead, the proposed search space (called neural channel expansion) is motivated by an insightful observation that the structure (not the neural network's parameter values) directly impacts the dynamic range of activation, resulting in reduced quantization error. As concrete evidence, we added Fig. 5 to show that NCE can increase SQNR, which leads to improved model accuracy.
>
> - NCE achieves significant accuracy gain with reduced hardware cost for 2-bit quantization, especially with the model of sufficient size (e.g., VGG16-CIFAR10, ResNet50-ImageNet). We want to emphasize that NCE even outperforms the SOTA mixed-precision model (EdMIPS, CVPR2020), as shown in Table 2.
>
>
> Q) LSQ seems to already do very well and it also uses uniform quantization.
>
> A) The reviewer is correct that LSQ achieved SOTA accuracy with uniform quantization. However, we would like to point out that LSQ's contributions (such as step-size gradient and scaling) are complementary to our work, and NCE (with PACT) already shows excellent potential even without them (this is particularly true for ResNet50-ImageNet). Note that NCE finds the channel structure adapted for the given quantizer, improving its base accuracy. In this work, we demonstrated this concept using PACT, but we are interested in trying out other SOTA quantizers as future work.
>
>
> Q) One major issue I have with the comparison in Tables 1 and 2 is that on some of the smaller networks the NCE result has more params than the other methods. ... I think you should uniformly increase channel widths in at least the "w/o NCE" baseline to see if the accuracy boost is from NCE learning the layer sensitivities or just from a bigger model.
>
> A) We agree with the reviewer that NCE seems to be more effective with the large models. However, we confirmed that NCE's benefit does not originate from simple channel expansion. To support our claim, we compare NCE with two existing channel expansion strategies: uniform increase of channels (WRPN) and pruning of channels from the enlarged model (TAS). As shown in Fig. 4 and Fig. 3c, respectively, NCE outperforms both WRPN and TAS approaches in accuracy vs. hardware-cost trade-off.
>
> A) (cont'd) We would also like to mention that we can control the increase of PARAM by simple tuning of a hyperparameter: the maximum number of channels. As a concrete example, we performed an ablation study in Table 3, where the maximum number of channels was set as the multiple of the original network during NCE search. Interestingly, the 1.5X case achieved satisfying accuracy with lower PARAM than the 2.0X case (which is the setting we reported in Table 2).  This observation implies that NCE can find a compelling network structure even with the more restricted search space.
>
>
> Q) I would also like to see exactly which layers were reduced in size on these networks.
>
> A) Thanks to the reviewer's suggestion, we presented the structure comparison for ResNet50-ImageNet in Fig. 6. As explained in detail in Sec. 7.2.5, we can see an interesting trend that the original structure artificially employs a vast number of channels in the later layers of ResNet50. In contrast, NCE effectively balances the channels across the layers, achieving better accuracy with uniform quantization.
>
>
> Q) The authors should include a comparison against such methods (NAS techniques for mixed-precision quantization) or discuss why it isn't needed.
>
> A) We did have a comparison of accuracy with a SOTA NAS-based mixed-precision method (EdMIPS, CVPR2020). As shown in Table 2, NCE outperforms EdMIPS for both ResNet18 and ResNet50 on the ImageNet dataset. We believe that this shows great potential for NCE.
>
>
> Q) Typos and errors
>
> A) Thanks for pointing them out. We will further proofread the text carefully to remove all the errors.

---

### Author Response · Authors · 2020-11-24
**General statement to the reviewers**

We sincerely thank the reviewers for the constructive comments and suggestions. We revised the main text (in blue) and added an in-depth ablation study in Appendix (Sec. 7) to address all the issues raised. In particular, a summary of materials added in Appendix is as follows:
- Table 3: Impact of constraining the maximum number of channels for NCE search. [Reviewer 4]
- Table 4: Comparison of quantization performance between NCE and 2X TAS with controlled search space size. [Reviewer 1,3,4]
- Figure 4: Comparison of quantization performance between NCE and WRPN with respect to PARAM and FLOPs on ResNet20-CIFAR10. [Reviewer 1,3,4]
- Figure 5: Relationship between activation dynamic range and quantization error. [Reviewer 1,3,4]
- Figure 6: Comparison of the structure of ResNet50-ImageNet before and after adaptation by NCE. [Reviewer 4]
- Table 5: Ablation study for threshold. [Reviewer 1]
- Table 6: Quantization with higher bit-precision for NCE. [Reviewer 1]

---

### Decision · Program_Chairs · 2021-01-07
**Final Decision**

**Decision:**

Reject

**Comment:**

This paper suggests a NAS approach for quantization which focuses on expanding the number of channels in problematic layers, given some uniform quantization level for all the layers. The reviewers were initially all negative, but the authors added more experiments and the scores changed to borderline (6/6/5). I think that though the reviewers appreciated the effort made by the authors and clarifying many of the issues. Yet, I think two concerns still remained. First, the method is demonstrated convincingly in only one large scale case (ImageNet) - on ResNet50 W2A2. The other case (ResNet18) is less impressive since it uses more parameters and has a small gap from EdMIPS. I think using more architectures (and possibly also precision levels) is important to convincingly demonstrate the utility of the method (e.g. EdMIPS used 5 architectures). Second, the novelty of the method in comparison to the previous method was not completely clear. Though the authors added more experiments on CIFAR to compare with previous methods, the significance of the results is not clear (small differences in the figures, and no error bars), and also the explanation of why the results of NCE should be better than TAS.